# Widely Targeted Metabolomics Was Used to Reveal the Differences between Non-Volatile Compounds in Different Wines and Their Associations with Sensory Properties

**DOI:** 10.3390/foods12020290

**Published:** 2023-01-08

**Authors:** Weiyu Cao, Nan Shu, Jinli Wen, Yiming Yang, Yanli Wang, Wenpeng Lu

**Affiliations:** Institute of Special Animal and Plant Sciences of Chinese Academy of Agricultural Sciences, Changchun 130117, China

**Keywords:** non-volatile metabolites, sensory evaluation, color difference analysis, weighted gene co-expression network analysis

## Abstract

In this study, metabolites from six varieties of wines, including ‘Haasan’ (A1), ‘Zuoshaner’ (A2), ‘Beibinghong’ (A3), ‘Shuanghong’ (A4), ‘Zijingganlu’ (A5), and ‘Cabernet Sauvignon’ (A6), were identified and quantified using widely targeted metabolomics analysis techniques. Based on the test results, 1172 metabolites were detected and classified into 18 categories. These include 62 amino acids, 178 alkaloids, 189 flavonoids, 106 phenols, 148 terpenoids, etc. Comparing the differential metabolites between the comparison groups of each variety, differences between varieties based on P-values and VIP values were shown. Among these differential metabolites, Trimethoprim and Crotonoside were screened out as core differential metabolites. Multiple comparisons also screened the biomarkers for each species. We used widely targeted metabolomics to reveal the differences between non-volatile compounds in different wines and their associations with sensory properties. We also used the simultaneous weighted gene co-expression network analysis (WGCNA) to correlate metabolites with sensory traits, including color difference values and taste characteristics. Two of the six key modules were screened by WGCNA for relevance to sensory traits (brown module and turquoise module). This study provides a high-throughput method for linking compounds to various sensory characteristics of food, opening up new avenues for explaining differences in different varieties of wine.

## 1. Introduction

With the development of society and the improvement of people’s living standards, more and more consumers are concerned about wine’s nutritional and health value, so consumer demand for wine is also increasing. At present, a research focus of the wine industry is to produce higher quality wine to meet the needs and desires of consumers to improve the scale of wine production and sales [1]. Wine has health benefits, and compounds in wine have been shown to have significant anti-inflammatory effects [2];, in vitro cellular experiments have demonstrated that red wine reduces ROS (reactive oxygen species, ROS) production, upregulates nicotinamide adenosine dinucleotide-dependent deacetylases SIRT1 and SIRT6 expression, protects circulatory system function, and prevents endothelial dysfunction [3]. Resveratrol in wine can reduce the frequency of cardiovascular morbidity, inhibit α-glucosidase activity, reduce glucose uptake by adipocytes, and has a hypolipidemic and fat deposition-inhibiting effect [4]. Phenolic substances derived from grapes have strong anti-tumor effects, including anti-invasive, pro-apoptotic, antioxidant, anti-proliferative, tumor cell cycle modulation, and angiogenesis inhibition [5]. Polyphenolic substances also positively affect cardiovascular disease prevention, neurodegenerative diseases, obesity prevention, cancer prevention, bone protection, and regulation of intestinal flora due to their various biological activities such as antioxidant, anticancer, and antibacterial [6,7]. A growing body of research focuses on the health benefits of wine.

In recent years, domestic wines have gradually entered the consumer’s field of vision. The main wine grape varieties grown in China are red and white. The red varieties are ‘Cabernet Sauvignon’, ‘Cabernet Franc’, ‘Merlot’, ‘Matheran’, etc. The most widely planted variety is ‘Cabernet Sauvignon’. ‘Cabernet Sauvignon’ originated in France. Its fruit has small thick skin, deep color, high acidity, and high tannin content, produces a wine with deep color and rich flavor, and is planted in Chinese wine regions. ‘Cabernet Franc’, ‘Merlot’, and ‘Matheran’ are made as single-varietal wines and often blended with other varieties to combine flavors and balance the structure. China’s main white grape varieties are ‘Chardonnay’, ‘Riesling’, and ‘Guinness’. ‘Chardonnay’ is the most adaptable and can show different flavors in different climatic cultivation conditions. Our endemic wine grape resource is the *Vitis amurensis*, one of the most cold-resistant species in the genus Vitis. The main varieties are ‘Zuoshanyi’, ‘Zuoshaner’, ‘Shuanghong’, ‘Shuangyou’, ‘Shuanfeng’, ‘Zuoyouhong’, ‘Beibinghong’, etc. Rich in anthocyanins and resveratrol, *Vitis amurensis* wines are black in colour and have a mellow, lightly grassy aroma [8]. There are many studies on metabolites in wine. However, little has been reported on the metabolomics of *Vitis amurensis*, a unique wine grape resource in China. The overall metabolism of wine and the metabolic profiles between different varieties of wine are not fully understood. Different varieties of wine are rich in different bioactive substances and have different nutritional values. Therefore, we used widely targeted metabolomics to explore the bioactive substances rich in different varieties of wine, the metabolic profiles of different varieties of wine as a whole, and the metabolite differences between individuals.

Nowadays, international research on wine grapes mainly focuses on European subspecies, European and American grape. The research on our unique wine grape *Vitis amurensis* is less, and research has certain limitations. In recent years, in particular, the impact of imported wines, coupled with changes in the domestic consumption environment, has led to sales of domestically produced wines hitting a snag, stocks building up, a decline in wineries’ willingness to make wine, a precipitous fall in wine consumption and a significant reduction in market size [9]. Increasing research into domestic wines is a top priority for today’s Chinese wine market.

With the increasing use of metabolomics in food analysis, many compounds in food can be quantified, and a high throughput method is needed to correlate compounds with various sensory characteristics of food to screen for compounds associated with sensory quality. In this study, basic physicochemical properties, chromatic hues, and widely targeted metabolomic analyses were performed on six different wine varieties, aiming to reveal sensory differences and related metabolite differences in wines made from different varieties and to reveal metabolites that may lead to differences between wines from different varieties, and to screen each variety for bioactive substances in order to find characteristic metabolites and biomarkers of wines from different varieties and the main differential metabolic pathways.

Our work provides valuable data for assessing the value of different wines, finding differences in the quality of different wines, and providing a theoretical basis for nutritional value assessment and food development. It also helps to identify the best varieties to guide future production strategies based on their use in the food processing industry, pharmaceuticals, plant breeding programs, etc.

## 2. Materials and Methods

### 2.1. Harvesting and Fermenting Method

On 25 September 2021, we took six varieties of grapes for testing, namely ‘Hassan’, ‘Zuoshaner’, ‘Beibinghong’, ‘Shuanghong’, ‘Zijingganlu’, and ‘Cabernet Sauvignon’, as shown in Table 1. The sampling site was the National fruit tree germplasm *Vitis amurensis* nursery in Zuojia town, Institute of Special Products, Chinese Academy of Agricultural Sciences (latitude 126°08′20″, longitude 44°06′82″). The sampling period is the fruit sampling period. Due to the short daylight and insufficient light, and temperature accumulation in the north-east, seed maturity is combined to determine whether the fruit is mature. During sampling, random samples were taken from different locations in the vineyard. The ears free of disease, insect pests, and moldy fruit were selected for artificial random sampling in the trees’ upper, middle, and lower parts of the trees, the shaded and exposed parts, and the two sides of the trees. Thirty kilograms of samples were collected for each variety for brewing the original wine. Samples were taken in a sampling bag, placed in a holding tank, and returned to the laboratory for fermentation the same day. The fermenters were selected from Tiburth 304 thermostatic fermenters. For fermentation (Figure 1), the grapes were destemmed, crushed, and fermented at room temperature (25 °C), CEC01 yeast from Angel’s Yeast was used for fermentation, and the amount of yeast added was 250 mg/Kg. The experiment was repeated three times for each wine sample. The first fermentation is seven days, the fermenter is closed during fermentation, and the exhaust valve is used to ensure that the gas generated in the fermentation process can be discharged smoothly. At the end of the first fermentation, the total sugar content of the wine stops falling, the indicators stabilize, and there are no bubbles in the fermenter. Then, we separate the fruit residue from the supernatant and ferment the supernatant for a second time. The second fermentation takes place for one month, after which the original wine is obtained, with the second fermentation taking place at a temperature of between 18–20 °C. After secondary fermentation, in November 2021, basic physicochemical properties testing, sensory evaluation, and extensive targeted metabolomic analysis of the different varieties of wine was conducted. 

### 2.2. Detection Methods of Basic Physical and Chemical Indexes

We measured soluble solids content with a handheld refractometer, and the titratable acid content of wine was determined by the indicator method according to GB/T 15,038–2006 General Analysis Method of Wine and Fruit Wine. The alcohol content was deter-mined by the alcohol meter method; Anthrone and sulfuric acid colorimetry were used to determine the total sugar content in grapes wine; the Folin–Denis reagent method was used to determine tannin content in grapes Juice; Measurement of anthocyanin content in wine by colorimetric method; Total phenol content: Folin–Ciocalteu colorimetric method [10]. Dry extraction content: Refer to the dry extraction test method of the national standard (GB/T 15,038–2006).

### 2.3. Color Difference Measurements of Wine Infusions

Colorimetric analysis spatially measures the color characteristics of wine samples according to the CIEL*a*b* colorimetric standard. Use the Lanbda 365 UV-Vis spectrophotometer to scan continuously (400–700 nm), distilled water as the blank control group, centrifuge the sample, filter it with a 0.22 µm filter head, and dilute the filtrate ten times to measure the absorbance at wavelengths of 450, 520, 570, and 630 nm, calculate the values of L*, a*, b*, Cab*, hab*, ΔEab* according to the four absorbances, the L value represents brightness, a* = red-green deviation, b* = blue-yellow deviation, hab* represents hue angle, Cab* represents the red wine color index, and ΔEab* represents the total color difference.
(ΔE_ab_*)^2^ = (L* − L_0_*)^2^ + (a* − a_0_*)^2^ + (b* − b_0_*)^2^(1)

### 2.4. Sensory Evaluation of Original Wine of Different Varieties

A quantitative descriptive analysis (QDA) of the wines was carried out by a trained sensory panel of 13 sommeliers (8 women and 5 men, aged from 22 to 52 years, average 34 years). These experts were recruited on the basis of their motivation and availability, and the panel members are all national tasting engineers, trained in accordance with national standards ISO 6658 and ISO 8586 prior to the sensory evaluation. The experts discussed the tasting properties of the wine in depth through three preliminary meetings (two hours each) until they all agreed on the degree of taste. According to the definition in published literature, according to the definition in national standard GB 15038-2006 and combined with references [11,12], in conjunction with the discussion results, the sensory attributes of wine taste were quantified using six sensory descriptors (Balance and Coodination, Thickness, Astringency, Aftertaste, Layering, and Acidity). Samples were labeled with three numbers and presented to the tasters randomly. Panel members were asked to rate the strength of each attribute on a 10-point scale, where a score of 10 indicated the highest strength and a score of 0 indicated the absence of it. Each sample was assessed in triplicate, and the mean of each sample was expressed by a triplicate average score based on a ten-point scale.

### 2.5. Widely-Targeted Metabolomic Analysis of Nonvolatile Metabolites in Raw Grape Wines

#### 2.5.1. Sample Preparation and Extraction

The metabolomic analysis was conducted by a commercial service company [13] according to a previously reported method [14].

The wine samples were stored at −80 °C until metabolite extraction. Frozen samples were freeze-dried under vacuum and ground using a mixer mill (MM400, Retsch, Shanghai, China) with a zirconia bead for 1.5 min at 30 Hz. Then, 100 mg powder was weighed and extracted overnight at 4 °C with 1.0 mL 70% aqueous methanol containing 0.1 mg/L lidocaine for the internal standard. Following centrifugation at 10,000× *g* for 10 min, the supernatant was absorbed and filtrated (SCAA-104, 0.22-μm pore size; ANPEL, Shanghai, China, www.anpel.com.cn/, accessed on 1 December 2021) before UHPLC-MS/MS analysis. 5 μL of each sample is mixed to prepare a quality Quality Control (QC). Quality Control (QC) samples to detect the reproducibility of the whole experiment.

#### 2.5.2. High-Performance Liquid Chromatography Conditions

UHPLC analysis conditions mainly include: (1) Chromatographic column: Waters ACQUITY UHPLC HSS T3 C18 1.8 μm, 2.1 mm × 100 mm. (2) Mobile phase: The aqueous phase is ultrapure water (added with 0.04% acetic acid), and the organic phase is acetonitrile (added with 0.04% acetic acid). (3) Elution ladder Degrees: Water: Acetonitrile, 95:5 *v*/*v* at 0 min, 5:95 *v*/*v* at 11.0 min, 5:95 *v*/*v* at 12.0 min, 95:5 *v*/*v* at 12.1 min, 15.0 min is 95:5 *v*/*v*. (4) The flow rate is 0.4 mL/min. (5) The column temperature is 40 °C. (6) The injection volume is 5 μL.

#### 2.5.3. ESI-QTrap-MS/MS Analysis

Samples were separated into ESI-QTRAP-MS for mass spectrometry analysis. The effluent was connected to an ESI-triple quadrupole-linear ion trap (QTRAP)–MS. Peak detection was performed using an ESI-triple quadrupole-linear ion trap (QTRAP, AB Sciex QTRAP6500 System, AB SCIEX Pet. Ltd., Framingham, MA, USA), with the following operating parameters: ESI source temperature 550 °C; ion spray voltage (IS) 5500 v; curtain gas (CUR) 25 psi; and collision-activated dissociation (CAD). Triple quadrupole (QQQ) scans were acquired as multiple reaction monitoring (MRM) experiments with optimized declustering potential (DP) and collision energy (CE) for each individual MRM transition. The scan range was set between 50 and 1000 *m*/*z*.

#### 2.5.4. Qualitative and Quantitative Metabolite Analyses

Data filtering, peak detection, alignment, and calculations were performed using Analyst 1.6.1 software. (AB SCIEX Pet. Ltd., Framingham, MA, USA). Metabolites were identified based on internal and public databases (MassBank, KNApSAck, HMDB, MoTo DB, and METLIN). Comparing the *m*/*z* values, the RT, and the fragmentation patterns with the standards.

#### 2.5.5. Kyoto Encyclopedia of Genes and Genomes (KEGG) Annotations and Metabolic Pathway Analyses of Differential Metabolites

KEGG is the primary public pathway-related database that includes genes and metabolites. Metabolites were mapped to the KEGG metabolic pathways for pathway analysis and enrichment analysis. Pathway enrichment analysis identified significantly enriched metabolic pathways or signal transduction pathways in differential metabolites compared to the background. The calculating formula is as follows:(2)$$P=1-\overset{m-1}{\underset{i=0}{\sum }}\frac{\left(\begin{array}{c} M\\ i \end{array}\right)\left(\begin{array}{c} N-M\\ n-i \end{array}\right)}{\left(\begin{array}{c} N\\ n \end{array}\right)}$$

Here, *N* is the number of all metabolites with KEGG annotation, *n* is the number of differential metabolites in *N*, *M* is the number of all metabolites annotated to specific pathways, and *m* is the number of differential metabolites in *M*. The calculated *p*-value went through FDR Correction, taking FDR ≤ 0.05 as a threshold. The *Pathways meeting* conditions were defined as significantly enriched pathways in differential metabolites.

### 2.6. Association of Non-Volatiles and Sensory Attributes via WGCNA

Sensory attributes, including three color difference values (L*, a* and b*) and six taste attributes (balance and harmony, thickness, astringency, finish, layering, and acidity), were studied for their correlation with non-volatiles. Co-expression networks were constructed using WGCNA (v1.47) package in R [15]. After filtering genes, gene expression values were imported into WGCNA to construct co-expression modules using the automatic network construction function blockwiseModules with default settings, except that the power is 6, TOMType is unsigned, and minModuleSize is 50. These modules are built using the DynamicTreeCut algorithm and assigned to different colors. Highly correlated metabolites are aggregated into a single module. Genes were clustered into 7 correlated modules. After calculating the topological overlap measure (TOM) using the adjacency matrix, the dissimilarity TOM was used to plot the dendrogram. The modules were established using the Dynamic Tree Cut algorithm and assigned to different colors. Highly correlated metabolites were clustered into one module.

### 2.7. Multivariate Data Analysis and Statistical Analysis

The fundamental physical and chemical indicators of the different varieties of wine were sorted in Excel 2010 and analyzed three times, with the results expressed as mean ± standard deviation (SD) in SPSS 23. Univariate analysis of variance (ANOVA) and the Duncan’s multi-interval test was used to determine the significance of the differences among the samples, and the significance level was 0.05. Radar plots were drawn using Excel 2010, using the mean scores for each attribute; a two-tailed Pearson correlation test was used to determine the correlation between the means of the different indicators, and hierarchical cluster analysis was used to group the different varieties of wine; SCIEX Analyst workstation software (version 1.6.3) for MRM data acquisition and processing; heat map is the use of metabonomics z-score value of the data set to draw heat maps, and free online platform in the OmicShare tool (http://www.omicshare.com/tools, accessed on 12 August 2022) to draft.

## 3. Results

### 3.1. Measurement Results of Fundamental Physical and Chemical Indexes of Wine

Using the same fermentation method to ferment six varieties of wine grapes, there were significant differences between the fundamental physicochemical indicators of the wine samples, as shown in Table 2. The total sugar and solids content of the wine determines the type of wine, which can be classified as dry, semi-dry, semi-sweet, or sweet, depending on the sugar content of the wine [16]. ‘Hassan’ had the highest solids content at 9.4 g/L, which differed significantly (*p* < 0.05) from the other five varieties of wine samples. Phenols are responsible for the bitterness and astringency and have a function in promoting wine quality (color, flavor, and mouthfeel), as well as being highly relevant in terms of health terms of antioxidants and cardioprotection [17,18]. The main coloring compounds in red wines are anthocyanins, which can be macerated from the grape skins and transformed during production and storage to form a system of anthocyanins and their derivatives, the composition and content of which affect the color characteristics of the wine [19]. ‘Zuoshaner’ had the highest contents of total acid, total sugar, total anthocyanin, and total phenols with 16.25 g/L, 5.7 g/L, 1477.85 mg/L, and 3.56 g/L, respectively, which were significantly different from other varieties (*p* < 0.05). Tannins are the source of bitterness and astringency in fruit wines. They are an essential component of the backbone of fruit wines, as well as having a very positive effect on stabilizing color, preventing oxidation, and removing off-flavors [20]. ‘Shuanghong’ had the highest tannin content of 3.64 g/L, significantly different from the other varieties (*p* < 0.05). The dry extract content is an essential indicator of the quality of wine, mainly determined by the variety and the age of the wine [21], and the analysis of the dry extract content in wine can tell whether the wine is adulterated with water, alcohol, etc. [22]. According to the national standard of China, the dry extract of red wine should not be less than 18.0 g/L. The above table shows that the dry extract content of the six wine varieties meets the national wine-making standard. At the end of the second fermentation, samples of raw wine from the six grape varieties ranged from 11 to 13 degrees alcohol.

### 3.2. CIELAB Parameters of Wine Samples of Different Varieties

The color of a wine is one of its essential organoleptic characteristics [23]. Parameter a* indicates the red-green color of the wine, b* the yellow-blue color of the wine, L* the lightness and darkness of the wine, Cab the saturation of the wine, Hab the hue angle of the wine, and ΔEab* the inter-wine color difference [24]. The calculated L*, a*, and b* CIELAB color spaces visually represent wine color characteristics [25,26]. As seen in Table 3, the L* values for the raw wine samples of all six varieties are high, ranging from 41.65 to 84.65, indicating that all six varieties have a good luster. The largest L* is ‘Cabernet Sauvignon’ at 84.65, the brightest colored wine; the minor L* is ‘Zuoshaner’ at 41.65, the darkest. The color a* value indicates the red hue of the wine, with a* value ranging from a maximum of 167.48 for ‘Zuoshaner’ to a minimum of 17.5 for ‘Cabernet Sauvignon’. Colour b* values, which indicate the yellow hue of the wine, ranged from 4.95 to 21.76 for wine samples, with the largest being ‘Zuoshaner’ and the smallest being ‘Cabernet Sauvignon’. Saturation cab* is a combination of a* and b*, indicating the color saturation of the wine color, the maximum saturation being ‘Zuoshaner’ and the minimum being ‘Cabernet Sauvignon’ 17.93, with wine samples having close saturation and chroma values, a characteristic of young wines. The color Angle of the wine samples is between 5.31 and 23.52, indicating that the color is close to purple color; the largest color Angle is ‘Zijingganlu’, indicating that the color is not so red compared with the other five varieties; the smallest is ‘Shuanghong’, indicating that the ‘Shuanghong’ wine is closest to purple color. The color difference between the ‘Cabernet Sauvignon’, which had the highest L* value, was used as a base value for the analysis of the six samples tested. It was found that there was a strong difference in color between the Cabernet Sauvignon and the other five samples. The color difference between ‘Cabernet Sauvignon’ and ‘Zuoshaner’ is the most pronounced. At the same time, according to Table 3, it can be noted that ‘Cabernet Sauvignon’ wines have a smaller red hue compared to other varietal wines, which is characteristic of Eurasian varietal wines, while domestic *Vitis amurensis* wines have a heavier red hue due to their rich phenolic content [27].

### 3.3. Sensory Scoring of Wine Samples

The sensory evaluation was done using six taste attributes (Balance and Coodination, Thickness, Astringency, Aftertaste, Layering, and Acidity) to evaluate the flavor of six wine samples. Statistical analysis showed that the six samples differed in each descriptor (Table 4, Figure 2). These significant differences indicate that the sensory intensity of each sample is significantly different. We did not find significant interactions between panel members and repetitions in our study, indicating that all panel members scored credibly on all descriptors. Further, the absence of significant interactions between the sample and replication and panel membership suggests that the sensory data are valid and credible. The results of the sensory evaluation revealed that, compared to the other samples, the ‘Cabernet Sauvignon’ has the best balance and harmony, the ‘Shuanghong’ has the best thickness, the ‘Zuoshaner’ has the best astringency and the highest acidity, the ‘Hassan’ has the best finish and the ‘Cabernet Sauvignon’ has the best layering.

### 3.4. Non-Volatile Metabolite Profiles of Samples of Different Varieties of Wine

#### 3.4.1. Widely Targeted Metabolome Analysis in Wine

In order to better understand the metabolite differences between different varietal wine varieties, we performed extensive targeted UHPLC-MS/MS metabolite analysis on six varietal wine samples.

A total of 1172 metabolites were detected and classified into 18 categories based on the test results. These include 62 amino acids, 178 alkaloids, 189 flavonoids, 106 phenols, 148 terpenoids, 25 quinones, 48 steroids, 23 lignans, 26 coumarins, 52 carboxylic acids and organic oxygen species, 15 vitamins and organic acids, 35 nucleotides, 42 phenylpropanoids, 27 sugars and alcohols, 73 lipids and aromatic species, 15 phytohormones, 22 tryptamines, indoles, pyridines, imidazoles, and their derivatives, etc. These include 62 amino acids, 178 alkaloids, 189 flavonoids, 106 phenols, 148 terpenoids, 25 quinones, 48 steroids, 23 lignans, 26 coumarins, 52 carboxylic acids and organic oxygen species, 15 vitamins and organic acids, 35 nucleotides, 42 phenylpropanoids, 27 sugars and alcohols, 73 lipids and aromatic species, 15 phytohormones, 22 tryptamines, indoles, pyridines, imidazoles, and their derivatives, etc.

#### 3.4.2. Multivariate Statistical Analysis of Metabolic Profile Differences

PCA analysis is used to characterize metabolomics under multidimensional data through several principal components and to reveal the internal structure of multiple variables through several principal components. Thus, utilizing PCA plots, we can observe the differences between the different groups. The unsupervised multidimensional statistical analysis method (PCA) was used to discriminate the magnitude of variability between groups of samples, between subgroups, and between samples within groups for the different wines (Figure 3A). The contribution of PC1 was 25.6% and that of PC2 was 13.5%, and the six groups of samples showed a clear separation trend on the two-dimensional graph, with no outliers and good clustering of samples of the same wine species. The PCA results could reflect that the metabolites differed significantly between the six groups of samples and were clearly distinguished from the others. In the PCA plots, the quality control (QC) samples, i.e., the mixture of wine samples, are projected onto the same area; some even overlap, indicating that they have similar metabolic profiles and that our analysis is stable and reproducible.

From the correlation analysis (Figure 3B), the samples clustered well with each other, and the correlation coefficient of the samples was r > 0.8, indicating that our analysis was reliably reproducible and had a high degree of confidence in the differences between wine samples of different varieties. To eliminate the effect of quantity on pattern recognition, we log10 transformed the peak areas of each metabolite and then performed a hierarchical cluster analysis (Figure 3C). The cluster heat map was also significantly divided into six groups so that the principal component analysis and cluster analysis showed that the six varietal wines had different metabolite profiles. The differential results indicated that genetic variation strongly influenced the metabolites of the different varietal wines.

### 3.5. Key Compounds Associated with Wine Flavor Differences

Orthogonal Least Partial Squares Discriminant Analysis (OPLS-DA) is an algorithm derived from PLS-DA [28]. In contrast to PLS-DA, OPLS-DA combines orthogonal signal correction (OSC) and PLS-DA methods and can decompose X-matrix information into two correlated types uncorrelated with Y. The relevant information is concentrated in the first predictive component by removing irrelevant differences, and subsequent model testing and differential metabolite screening are analyzed using the OPLS-DA results. The method is a multivariate statistical analysis with supervised pattern recognition, which effectively removes study-irrelevant effects to maximize inter-group differences in metabolites.

OPLS-DA can evaluate the classification effectiveness of the model using R2X, R2Y, Q2, and OPLS-DA score plots, and VIP (Variable Importance for the Projection) values to indicate the importance of variables (feature peaks) that explain the X dataset and the associated Y dataset. In all sets of sample comparisons, Q2 is more significant than 0.9, indicating that the model is excellent and works better than the PCA model. Permutation validation of OPLS-DA (*n* = 200, i.e., 200 permutation experiments) revealed that both R2 and Q2 of the original model were more significant than R2 and Q2 of the post-Y replacement model, indicating that the model predictions were reliable. Variables with VIP > 1 can be used as screening conditions for potential biomarkers.

To avoid false positive errors due to only one statistical analysis method on the face, we further screened for differential metabolites based on VIP ≥ 1 and *t*-test *p* < 0.05. The number of differential metabolites is shown in Figure 4. Sixty-three metabolites screened between A1 and A2 (31 up-regulated, 32 down-regulated), 66 metabolites screened between A1 and A3 (25 up-regulated, 41 down-regulated); 68 metabolites screened between A1 and A4 (45 up-regulated, 23 down-regulated); 69 metabolites screened between A1 and A5 (13 up-regulated, 56 down-regulated); 59 metabolites screened between A1 and A6 (13 up-regulated, 46 down-regulated). Fifty-six metabolites screened between A2 and A3 (26 up-regulated, 30 down-regulated); 60 metabolites screened between A2 and A4 (47 up-regulated, 13 down-regulated); 61 metabolites screened between A2 and A5 (17 up-regulated, 44 down-regulated); 50 metabolites screened between A2 and A6 (14 up-regulated, 36 down-regulated). Seventy metabolites screened between A3 and A4 (58 up-regulated, 12 down-regulated); 66 metabolites screened between A3 and A5 (18 up-regulated, 48 down-regulated); 45 metabolites screened between A3 and A6 (13 up-regulated, 32 down-regulated). Eight-five metabolites were screened between A4 and A5 (14 up-regulated and 71 down-regulated); 51 metabolites were screened between A5 and A6 (52 up-regulated and 29 down-regulated).

Compare the differential metabolites between comparison groups A1, A2, A3, A4, A5, and A6. Based on the *p* and VIP values showing the differences between the varieties, the Venn diagram (Figure 5A) shows that two core differential metabolites were screened, these being Trimethoprim and Crotonoside (Table 5), which are phenolic and alkaloid compounds, respectively, that may be the key compounds determining the quality of the wines and causing the differences in flavor in these six wines. Crotonoside has a wide range of biological activities, as well as numerous pharmacological effects, including anti-arrhythmic effects by regulating the sodium and calcium channels of ventricular myocytes and rich anti-tumor activities [29,30,31,32]. Trimethoprim is an antibacterial synergist [33] with an excellent antibacterial effect [34], which is collected by the pharmacopeia of many countries and is widely used in clinical treatment and animal husbandry, becoming one of the pillar products of China’s pharmaceutical industry.

As seen on the PCA plot (Figure 3A), A2 and A4 are clustered, and A3 and A6 are clustered, indicating similarity in their metabolite content and classes. We further selected groups A1, A4, A5, and A6 as the comparison groups and compared the differential metabolites of each group two by two, followed by screening the differential metabolites between the comparison groups A1, A4, A5, and A6, which can show the differences between the species based on *p*-values and VIP values, as shown in the Venn diagram (Figure 5B). A total of eight differential metabolites shared between the groups were screened. The eight shared differential metabolites are shown in Table 5. These eight compounds include amino acid compounds, phenolic compounds, nucleotides, alkaloids, etc., which may determine the quality of the wine. In a two-by-two comparison, Trimethoprim was the only phenolic compound obtained by a two-by-two comparison, which may be related to the antioxidant activity of the wine. Guanine, 2’-O-Methyladenosine, D-Glutamine, the nucleotide-based carboxylic acid organoids, and Carbendazim, a benzimidazole, also showed differences in the pairing of A1, A4, A5, and A6. L-Glutamic acid and L-Lysine are the shared differential metabolites in the comparison, and the amino acids are an essential part of the wine’s flavor and nutrition. As can be seen in Table 5, the levels of L-Glutamic acid and L-Lysine in the double red wine A4 were significantly higher (*p* < 0.05) than in several other varieties of wine. L-glutamic acid is widely used in food, daily necessities, pharmaceuticals, health products, feed, and many other fields as a food additive and in the medical industry [35,36]. L-lysine is mainly produced during fermentation [37] and is one of humans and animals’ eight essential amino acids [38]. Crotonoside was the only alkaloid in a two-by-two comparison. These eight compounds screened may be the key differential metabolites affecting the quality of these wines.

### 3.6. Identification of Characteristic Metabolites of Each Genotype

The characteristic metabolites of A1 were identified by multiple comparisons. A total of 19 characteristic metabolites were screened, including three amino acids, three flavonoids, two phenolic compounds, three nucleotides, two lipids, etc. The relatively high levels of Cianidanol, Trimethoprim, Enol-phenylpyruvate, 5’-Deoxyadenosine, Cordycepin, Phenethyl alcohol, Styrene, 6-Methoxymellein, 2-Picolinic acid, and Cyanidin 3-rutinoside compared to the other metabolites can be used as biomarkers for screening A1.

A total of 18 differential metabolites, including eight flavonoids, two lipids, two carboxylic acids, one alkaloid, two terpenoids, one amino acid, and one phenolic compound, were screened between A2 and the respective species comparisons; the content of flavonoids differed significantly between A2 and the respective comparison groups. Astragalin, Cynaroside, Quercetin, Peonidin-3-glucoside, Quercetin-3-O-glucuronide, Acevaltrate, L-Acetylcarnitine, Leukotriene A4, and Kaurenoic acid are higher in content compared to other metabolites and can be used as biomarkers for screening A2.

A3 and comparison across species, a total of 12 differential metabolites were screened, including five flavonoids, two alkaloids, two terpenoids, one phytohormone, one phenol, and one organic acid. There was a significant difference in flavonoid content between A3 and the comparator groups, and three flavonoids, Astilbin, Cianidanol, and Cyanidin 3-rutinoside, were present at higher levels compared to the other metabolites, which could be used as biomarkers for screening A3.

A4 and comparison across species, 19 differential metabolites were screened, including five flavonoids, two amino acids, two nucleotides, two terpenoids, one phenol, two vitamins, and one alkaloid. Compound levels varied significantly between A4 and the comparator groups. The 15 different metabolites were significantly higher than several other varieties, except for L-Epicatechin, Chalconaringenin, Trimethoprim, and Leukotriene A4, which were lower than the other varieties. Notably, the Crotonoside content in A4 is exponentially higher than in the wines of the other varieties, which could be used as a biomarker for screening A4.

A5 and a comparison of the species, a total of 20 differential metabolites were screened, including two flavonoids, three amino acids, three alkaloids, three carboxylic acids and one phenolic. Four compounds, Cianidanol, Trimethoprim, Deethylatrazine, and Kynurenic acid, were present at higher levels than other metabolites, and these metabolites could be used as biomarkers for screening A5.

A6 and the species were compared, and 13 differential metabolites were screened, including two flavonoids, two amino acids, two phenols, two carboxylic acids, one nucleotide, one alkaloid, one lipid, one terpenoid, and one imidazole. Regarding differential metabolites between A6 and the comparator group of each variety, A6 had lower levels of each metabolite than the metabolites of the remaining five varieties, and no characteristic biomarkers were screened.

### 3.7. Non-Volatile Metabolites in Wine Linked to Important Organoleptic Traits through WGCNA

Before WGCNA analysis, the selected gene set was screened and filtered to remove low-quality genes or samples that caused instability in its results and to improve the accuracy of the network construction. Metabolites with low fluctuations in expression variation were filtered, leaving a final set of 1172 metabolites. Then, the network construction parameters were set. The gene co-expression network is a scale-free weight gene network, and the scale-free property means that the network’s degree distribution meets the power-law distribution. The distribution mainly presents a centralized and convergent distribution. In the state of distributed distribution, most nodes have a few connections, and a large number of connections are only a few nodes. In order to meet the premise of scale-free network distribution as much as possible, as shown in Figure 6, we set the power value as six in the network construction parameters, which can ensure a significant degree of data connectivity.

We built a weighted co-expression gene network based on the selected power values and divided the 1172 metabolites into seven modules. Gene clustering trees were constructed based on the correlation of gene expression, and gene modules were classified according to the clustering relationships between genes. Genes with similar expression patterns would be classified into the same module, and branches of the clustering tree were sheared to differentiate them, producing different modules, with each color representing a module and grey indicating genes that could not be classified into any one module. After the initial module division, the results of the initial module division were obtained as Dynamic Tree Cut (Figure 6A). Since some modules were very similar, we then merged modules with similar expression patterns based on the similarity of module eigenvalues to obtain the final module division as merged dynamic (Figure 6B). The similarity chosen for the analysis was 0.75. The minimum number of genes chosen for the analysis was 50.

A co-expression network was constructed using WGCNA to link 1172 metabolites with sensory characteristics, including color difference values (L*, A*, and b*) and taste attributes (balance, thickness, astringency, thickness, hierarchy, and acidity) and a total of seven co-expression modules were obtained based on their expression patterns, which were clustered into two main branches with opposite correlation patterns. The correlation coefficients are shown in Figure 7. The correlations and corresponding *p*-values are presented numerically at the intersection of the module and the physiological indicators. According to the correlation analysis of “module-trait”, the blue module is related to wine acidity (*p* < 0.05, r = 0.49), the brown module is related to wine a value (*p* < 0.05, r = 0.91), thickness (*p* < 0.05, r = 0.61), astringency (*p* < 0.05, r = 0.59) and acidity (*p* < 0.05, r = 0.62). The turquoise module is related to thickness (*p* < 0.05, r= 0.61), astringency (*p* < 0.05, r = 0.57), and acidity (*p* < 0.05, r = 0.54) in wine. Green module and b showed opposite correlation (*p* < 0.05, r = −0.6), brown module and L value (*p* < 0.05, r = −0.79), balance and coordination (*p* < 0.05, r = −0.65), aftertaste (*p* < 0.05, r = −0.74), and layering (*p* < 0.05, r = −0.76) showed an inverse correlation. The turquoise module was inversely correlated with the balance and coordination (*p* < 0.05, r = −0.85), the aftertaste (*p* < 0.05, r = −0.64), and the gradation (*p* < 0.05, r = −0.56) of the wine. When correlating the modules with the phenotype, it is clear that the brown and turquoise modules are the key modules and can be the main ones we will study. Firstly, the 202 metabolites in our brown module can be divided into a dozen categories, but most are flavonoids, amino acids, phenols, and lipids. The 321 metabolites in the turquoise module can be divided into a dozen categories, primarily amino acids, flavonoids, lipids, aromatic compounds, and alkaloids.

From the metabolites contained in each module (Figure 8A), the maximum number of metabolites gathered in the turquoise module was 321, while the number of metabolites gathered in the red module was 101. As shown in Figure 8B–G, metabolites in the blue module are highly expressed in A1, those in the brown module are highly expressed in A2 and A4, those in the green module are highly expressed in A1 and A4, those in the red module are highly expressed in A5, and those in turquoise module are highly expressed in A2 and A4. The metabolites in the yellow module are highly expressed in A2, A4, and A6. The metabolites in each color module are expressed in the corresponding groups and are associated with their corresponding sensory characteristic properties.

### 3.8. Classification and Enrichment Analysis of the Metabolite KEGG in the Key Module

We mapped 202 metabolites from the brown module to the KEGG database, and the 20 metabolic pathways with the lowest q-values are shown in Figure 9A, starting with a look at the information about the pathways. Our results show that most metabolites map to secondary metabolite ‘metabolic pathways’, as expected. A few metabolites were categorized as ‘genetic information processing’, ‘environmental information processing’, and ‘human disease’, suggesting that some metabolites may have potential health effects. The KEGG results show that “diterpenoid biosynthesis” and “anthocyanin biosynthesis” play an important role and biological function in the sensory properties associated with the brown module. The 321 metabolites in the turquoise module were mapped to the KEGG database, and the 20 metabolic pathways with the lowest q-values are shown in Figure 9B. It was found that most metabolites were mapped to ‘metabolic pathways’, and a few were classified as ‘genetic information processing’, ‘environmental information processing’, and ‘human diseases’, suggesting that some metabolites in the turquoise module also have potential health effects. The KEGG results also indicate that related metabolic pathways such as “aminyl tRNA biosynthesis”, “nicotinic acid and nicotinamide metabolism”, “alanine, aspartate, and glutamate metabolism”, “ABC transporter protein”, “amino acid biosynthesis”, “carotenoid biosynthesis”, and “glycine, serine and threonine metabolism” play an important role and biological function in the sensory properties associated with the turquoise module.

## 4. Discussion

Previous research has focused on the aromatic characteristics of wines, their main chemical components, and the factors that influence their flavor, i.e., the study of the flavor of wines. Few studies have reported on the differences in metabolites between wines of different varieties. In this study, we focused on the physiological indicators and metabolic differences between the different varieties of wine. In addition, we provide an in-depth and comprehensive interpretation of the flavor quality and metabolic profile characteristics of six wine varieties and establish preliminary metabolic markers for six varieties of wine, including five for *Vitis amurensis* wines. This study provides a theoretical reference for the targeted market application and promotion of wines.

Compared to previous studies, our results identify a greater variety of metabolites in wine and contain a greater number of metabolites. A total of 1172 metabolites were detected in different varieties of wine. Most metabolites map onto secondary metabolite ‘metabolic pathways’, including ‘lipid metabolism’, ‘carbohydrate metabolism’, ‘amino acid metabolism’, ‘nucleotide metabolism’, ‘flavonoid biosynthesis’ and other critical metabolic pathways that affect the flavor of wine, in addition to those associated with ‘human health’. This study also reports and analyses, for the first time, metabolites in the Chinese specialty variety *Vitis amurensis* wine.

Omics technology is widely used in the wine. Stefanou A et al. analyzed the metabolites of different grape varieties utilizing NMR and HPLC-MS instruments and multivariate statistical methods to identify the resistance of different varieties to Botis and powdery mildew. Lee Jang-Eun et al. [39] used metabolomics techniques to assess the impact of grape vintage on wine. Pereira Giuliano E et al. [40] used 1H NMR to identify metabolites in grapes and used multivariate statistical methods to geographically classify grapes from four Bordeaux appellations, providing a technical guarantee for the control of regional raw materials, as well as an idea and method for the identification of regional wines; metabolomics has been intensively studied in wine, and it is common to use metabolomics to distinguish between origin, varietal, and vintage wines. In the production and commercialization of wine, sensory evaluation is one of the most intuitive ways to assess the quality of a wine and the satisfaction it brings to the consumer. Extensively targeted metabolomics is an emerging histological technique that qualitatively and quantitatively analyses all low molecular weight metabolites in biological samples and screens for biologically significant differences in metabolites between groups to elucidate the mechanisms of metabolic processes and physiopathological changes in organisms.

This study uses broadly targeted metabolomics to detect non-volatile metabolites in different varieties of wine and correlate them with sensory characteristics, including color and taste attributes. A total of 2 expression modules related to sensory traits were obtained from the key modules by WGCNA analysis, with 523 metabolites related to sensory traits. This study identifies the relationship between the sensory characteristics of different varieties of wine and their non-volatile metabolites, providing a scientific basis for future wine quality improvement efforts.

## Figures and Tables

**Figure 1 foods-12-00290-f001:**
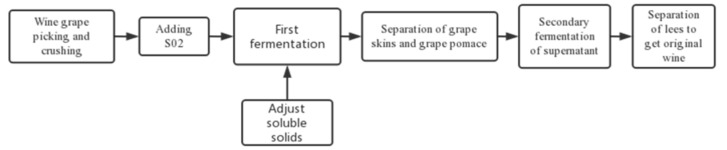
Fermentation experiment flow chart.

**Figure 2 foods-12-00290-f002:**
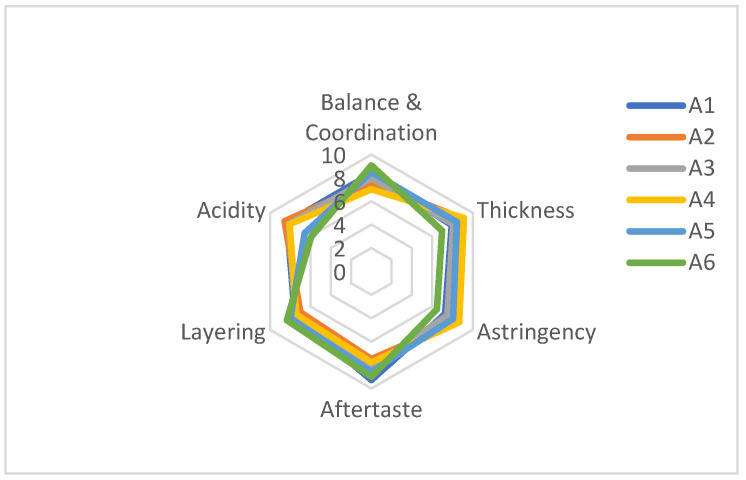
Radar chart for sensory evaluation of different varieties of wine. Note: A1—‘Hassan’; A2—‘Zuoshaner’; A3—‘Beibinghong’; A4—‘Shuanghong’; A5—‘Zijingganlu’; A6—‘Cabernet Sauvignon’.

**Figure 3 foods-12-00290-f003:**
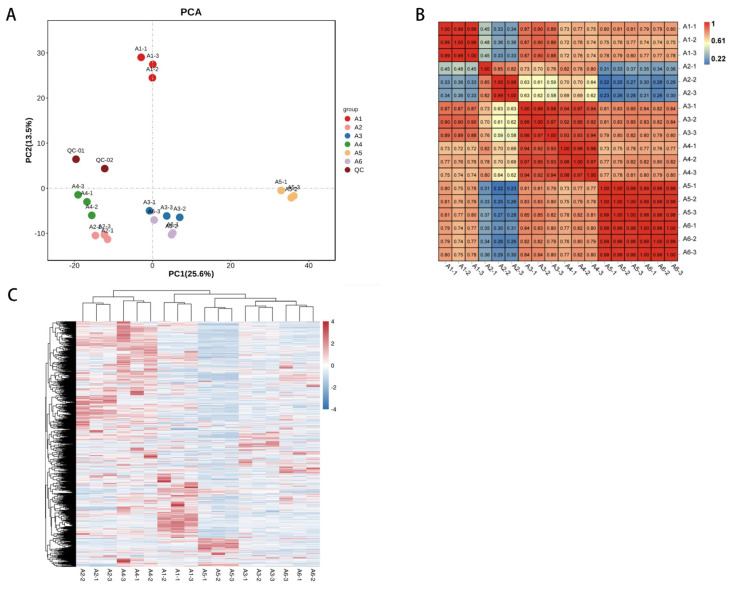
Principal component analysis, heat map analysis, correlation analysis of relative differences in the metabolite content of different varieties of wine. (**A**) Principal component analysis of the metabolites of six samples. Equal wine samples were mixed and used as quality control (QC). (**B**) Cluster heat map analysis of the metabolites of six varieties of wine samples. The colors indicate the accumulation level of each metabolite, with higher values from low (blue) to high (red) and darker colors indicating a stronger correlation between the two samples. (**C**) Cluster heat map analysis of the metabolites of six varieties of wine samples. The colors indicate the accumulation level of each metabolite, with higher values from low (blue) to high (red) and darker colors indicating a stronger correlation between the two samples.

**Figure 4 foods-12-00290-f004:**
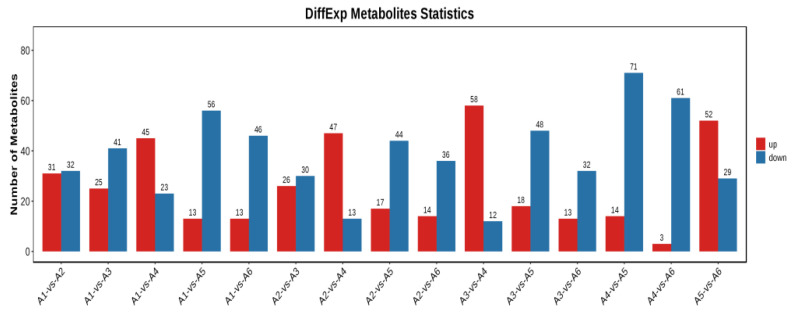
Statistical graph of the number of differential metabolites between varieties. The horizontal coordinates are for each comparison group, the vertical coordinates are for the number of differential metabolites in each comparison group, red represents the number of differential metabolites up-regulated in the latter relative to the former in the comparison group, and blue represents the number of differential metabolites down-regulated.

**Figure 5 foods-12-00290-f005:**
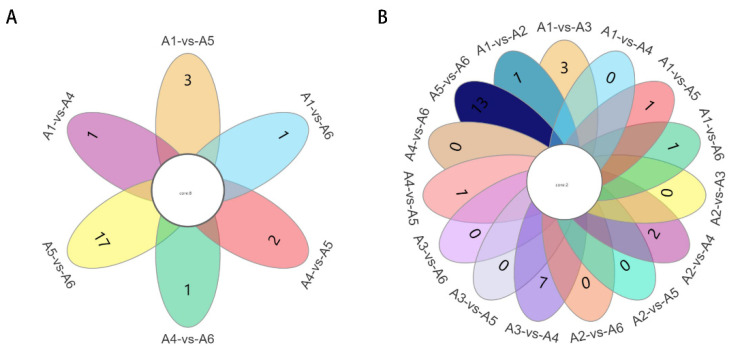
Paired comparison plots of differential metabolites (**A**) Venn diagram of differential metabolites for each comparison group of A1, A2, A3, A4, A5, and A6. (**B**) Venn diagram of differential metabolites for each comparison group of A1, A4, A5, and A6.

**Figure 6 foods-12-00290-f006:**
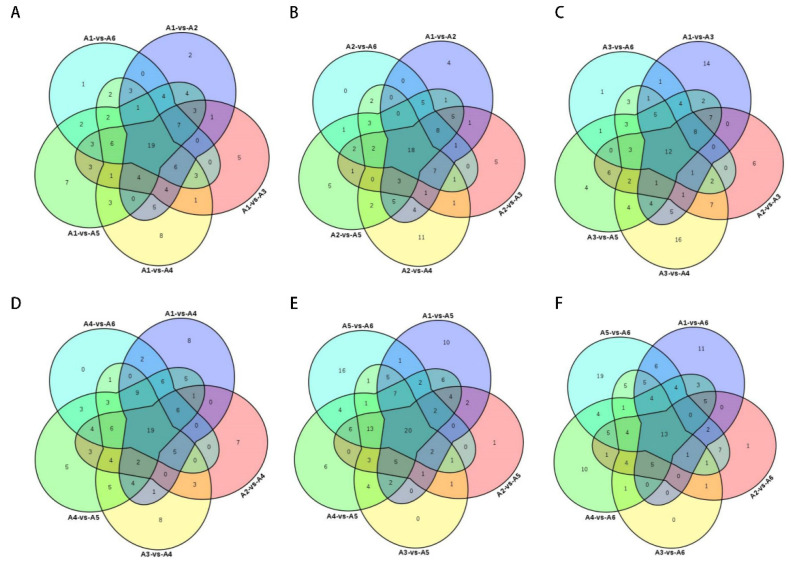
Venn diagram analysis of two-by-two differential metabolites with the other five groups, using groups A1, A2, A3, A4, A5, and A6 as core comparison groups, respectively. (**A**) Venn diagram analysis was performed with A1 as the core comparison group with the other 5 groups of differential metabolites. (**B**) Venn diagram analysis was performed with A2 as the core comparison group with the other 5 groups of differential metabolites. (**C**) Venn diagram analysis was performed with A3 as the core comparison group with the other 5 groups of differential metabolites. (**D**) Venn diagram analysis was performed with A4 as the core comparison group with the other 5 groups of differential metabolites. (**E**) Venn diagram analysis was performed with A5 as the core comparison group with the other 5 groups of differential metabolites. (**F**) Venn diagram analysis was performed with A6 as the core comparison group with the other 5 groups of differential metabolites.

**Figure 7 foods-12-00290-f007:**
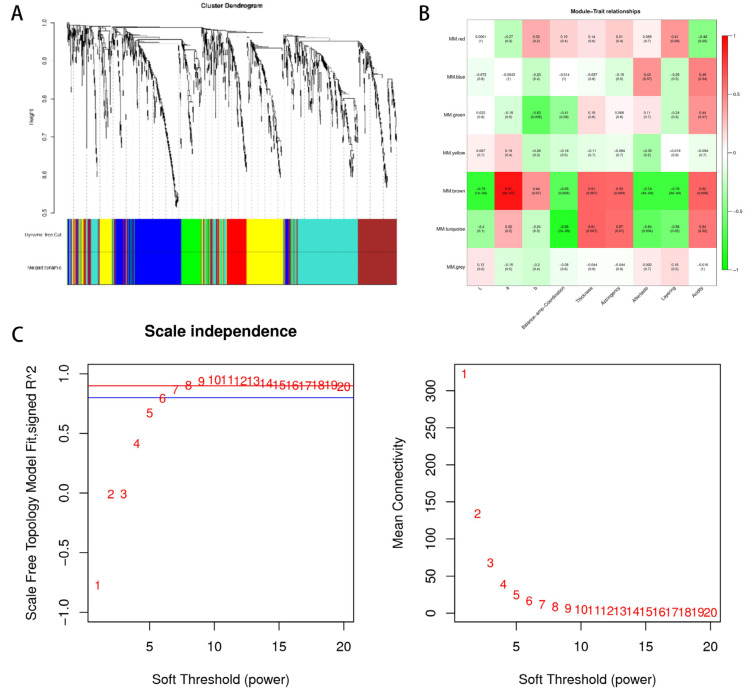
Correlations of metabolites with taste attributes based on WGCNA. (**A**) Clustering dendrogram of the average network adjacency for identifying metabolite co-expression modules. (**B**) Module-Trait Relationship is plotted with the trait in the horizontal coordinate and the module in the vertical coordinate, using the Pearson correlation coefficient. The number in the lower brackets represents the significance *p* value. The smaller the value, the stronger the significance. The graph reflects the correlation between each module and each trait. (**C**) Power value curves where the vertical distance represents the distance between two nodes (between genes) and the horizontal distance is meaningless.

**Figure 8 foods-12-00290-f008:**
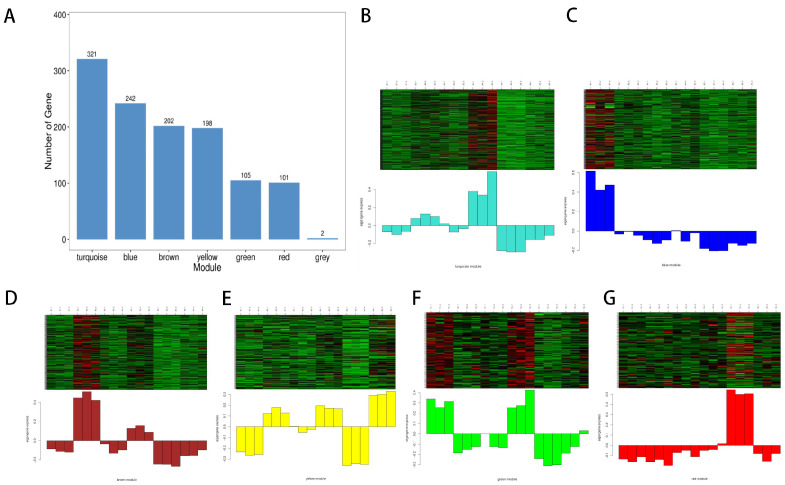
Heat map of metabolite numbers and expression by module. (**A**) Histogram of the number of genes in each module, with the horizontal coordinate indicating each module and the vertical coordinate indicating the number of genes. (**B**–**G**) Heat map of gene expression patterns by module, the expression patterns of each gene contained in each module are presented in a heat map, and the changes in module characteristic values across samples (equivalent to module expression patterns) are presented in a bar chart. In the figure, the top half is the heat map of the expression of genes in the modules in different samples, with up-regulation in red and down-regulation in green; the bottom half is the module characteristic values in different samples.

**Figure 9 foods-12-00290-f009:**
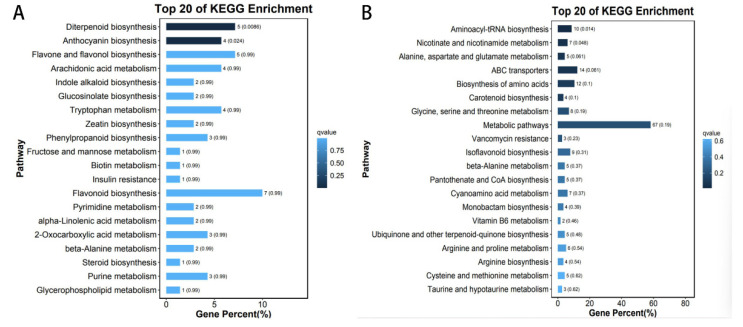
KO enrichment bar graph: (**A**) The 20 metabolic pathways with the lowest q-values in the brown module. (**B**) The 20 metabolic pathways with the lowest q-values in the turquoise module. The top 20 pathways with the smallest q-value are used to make the graph. The vertical coordinate is the Pathway. The horizontal coordinate is the number of that Pathway as a percentage of the number of all differences. The darker the color, the smaller the q-value, and the value on the bar is the number q-value of that Pathway.

**Table 1 foods-12-00290-t001:** Materials used in this study.

Varieties	Flower Type	Species	Parentage	Place of Origin
‘Hassan’	Bisexual Flower	*Vitis amurensis*	Wild species	China
‘Zuoshaner’	Female Flower	*Vitis amurensis*	Wild species	China
‘Beibinghong’	Bisexual Flower	*Vitis amurensis × Vitis vinifera*	‘Zuoyouhong’ × ‘84–26-53′	China
‘Shuanghong’	Bisexual Flower	*Vitis amurensis*	‘Tonghuasanhao’ × ‘Shuangqing’	China
‘Zijingganlu’	Bisexual Flower	*Vitis amurensis*	‘Zuoshaner’ × ‘Hassan’	China
‘Cabernet Sauvignon’	Bisexual Flower	*Vitis vinifera*	‘Cabernet Franc’ × ‘Sauvignon Blanc’	France

**Table 2 foods-12-00290-t002:** The basic physical and chemical indicators of the original wine.

Varieties	Solids (Brix%)	Titratable Acid/g·L^−1^	Total Sugar/g·L^−1^	Total Anthocyanins/mg·L^−1^	Total Phenols (g/L)	Tannin/g·L^−1^	pH Value	Dry Extract/g·L^−1^	Alcohol (*v*/*v*)
‘Hassan’	9.40 ± 0.00 ^a^	11.00 ± 0.29 ^c^	4.10 ± 0.12 ^c^	151.96 ± 6.68 ^e^	1.61 ± 0.04 ^d^	2.33 ± 0.31 ^c^	4.13 ± 0.01 ^b^	31.50 ± 0.00 ^d^	13.00 ± 0.00 ^a^
‘Zuoshaner’	9.17 ± 0.06 ^b^	16.25 ± 0.29 ^a^	5.70 ± 0.35 ^a^	1477.85 ± 48.31 ^a^	3.56 ± 0.04 ^a^	3.59 ± 0.07 ^b^	3.65 ± 0.01 ^f^	40.80 ± 0.17 ^a^	11.00 ± 0.00 ^c^
‘Beibinghong’	8.53 ± 0.06 ^d^	12.69 ± 0.43 ^b^	2.90 ± 0.24 ^d^	236.57 ± 14.97 ^d^	1.39 ± 0.03 ^e^	1.76 ± 0.05 ^d^	3.79 ± 0.01 ^e^	30.90 ± 0.17 ^e^	12.00 ± 0.00 ^b^
‘Shuanghong’	8.67 ± 0.58 ^c^	12.75 ± 0.00 ^b^	4.46 ± 0.21 ^b^	1440.56 ± 75.15 ^b^	3.00 ± 0.24 ^b^	3.64 ± 0.09 ^a^	3.84 ± 0.01 ^d^	37.13 ± 0.12 ^c^	13.00 ± 0.00 ^a^
‘Zijingganlu’	7.9 ± 0.00 ^f^	7.38 ± 0.29 ^e^	4.10 ± 0.12 ^c^	496.51 ± 10.74 ^c^	1.90 ± 0.10 ^c^	1.67 ± 0.10 ^e^	4.12 ± 0.01 ^c^	37.40 ± 0.17 ^b^	12.00 ± 0.00 ^b^
‘Cabernet Sauvignon’	8.1 ± 0.00 ^e^	8.25 ± 0.19 ^d^	1.65 ± 0.07 ^e^	108.54 ± 7.28 ^f^	0.82 ± 0.08 ^f^	1.36 ± 0.34 ^f^	4.39 ± 0.01 ^a^	27.30 ± 0.17 ^f^	12.00 ± 0.00 ^b^

Means with different letters in the same column express significant differences (Duncan’s test *p* < 0.05).

**Table 3 foods-12-00290-t003:** Analysis of color differences in different varieties of wine.

Varieties	L*	a*	b*	Cab*	hab*	ΔEab*
‘Hassan’	66.91 ± 0.06 ^c^	40.15 ± 0.01 ^c^	7 ± 0.28 ^d^	40.75 ± 0.21 ^d^	9.89 ± 0.48 ^d^	29.16 ^d^
‘Zuoshaner’	41.65 ± 0.10 ^f^	167.48 ± 0.07 ^a^	21.76 ± 0.17 ^a^	168.89 ± 0.10 ^a^	7.4 ± 0.77 ^e^	157.30 ^a^
‘Beibinghong’	73.76 ± 0.12 ^b^	35.46 ± 0.07 ^e^	7.24 ± 0.30 ^c^	36.2 ± 0.14 ^e^	11.54 ± 0.28 ^c^	21.46 ^e^
‘Shuanghong’	59.65 ± 0.1 ^e^	53.3 ± 0.26 ^b^	4.95 ± 0.07 ^f^	53.53 ± 0.24 ^b^	5.31 ± 0.27 ^f^	44.04 ^b^
‘Zijingganlu’	63.22 ± 0.11 ^d^	39.84 ± 0.07 ^d^	17.34 ± 0.28 ^b^	43.46 ± 0.19 ^c^	23.52 ± 0.4 ^a^	33.43 ^c^
‘Cabernet Sauvignon’	84.65 ± 0.12 ^a^	17.05 ± 0.14 ^f^	5.57 ± 0.11 ^e^	17.93 ± 0.11 ^f^	18.09 ± 0.25 ^b^	

Means with different letters in the same column express significant differences (Duncan’s test *p* < 0.05).

**Table 4 foods-12-00290-t004:** Sensory evaluation of different varieties of wine.

Taste	Total	‘Hassan’	‘Zuoshaner’	‘Beibinghong’	‘Shuanghong’	‘Zijinggsnlu’	‘Cabernet Sauvignon’
Balance & Coordination	10	8.37 ± 0.12	7.63 ± 0.06	8.00 ± 0.00	7.03 ± 0.06	8.50 ± 0.00	9.07 ± 0.12
Thickness	10	7.93 ± 0.06	9.00 ± 0.00	8.07 ± 0.12	9.13 ± 0.12	8.43 ± 0.12	6.97 ± 0.06
Astringency	10	7.27 ± 0.06	8.53 ± 0.06	7.50 ± 0.00	8.67 ± 0.06	8.07 ± 0.12	6.47 ± 0.06
Aftertaste	10	9.27 ± 0.25	7.50 ± 0.00	8.17 ± 0.29	7.77 ± 0.25	8.57 ± 0.12	9.00 ± 0.00
Layering	10	7.53 ± 0.15	7.00 ± 0.00	7.47 ± 0.06	7.40 ± 0.10	8.00 ± 0.00	8.33 ± 0.29
Acidity	10	8.33 ± 0.06	8.57 ± 0.12	8.13 ± 0.15	8.07 ± 0.06	6.57 ± 0.12	5.93 ± 0.12

**Table 5 foods-12-00290-t005:** Identification of differentially expressed metabolites from a two-by-two comparison of six wines.

Category	Compounds	A1	A2	A3	A4	A5	A6
		A1, A4, A5, A6 Two-by-Two Comparison of Shared Differential Metabolites					
Amino acids	L-Glutamic acid	8,740,339.55 ± 1,304,003.80	22,283,076.63 ± 1,298,588.33	19,698,562.46 ± 1,284,131.33	32,662,899.86 ± 7,929,513.51	844,119.76 ± 205,156.2	3,839,362.14 ± 1,319,648.25
Amino acids	L-Lysine; L-Glutamine	15,534,750.58 ± 1,117,706.63	22,766,130.05 ± 1,690,608.93	25,430,043.14 ± 2,134,330.58	38,305,543.74 ± 12,664,939.83	5,927,226.67 ± 815,294.35	10,829,465.34 ± 1,061,020.55
Phenols	Trimethoprim	170,392,569.41 ± 7,239,534.72	36,939,619.52 ± 884,872.69	91,544,508.75 ± 7,804,644.7	58,070,842.15 ± 469,344.95	131,185,043.54 ± 10,643,889.16	17,080,836.34 ± 527,750.76
Benzimidazoles	Carbendazim	129,093,442.01 ± 5,991,428.08	155,489,246.64 ± 12,991,581.68	162,246,727.33 ± 1,609,192.67	147,606,817.84 ± 6,560,923.58	59,830,134.44 ± 6,447,283.36	125,099.58 ± 11,249.83
Nucleotides	Guanine	283,570.58 ± 49,403.51	11,399,025.44 ± 7,981,916.55	22,148,403.38 ± 2,181,389.92	47,846,898.97 ± 4,624,801.58	23,971,289.84 ± 4,091,302.17	9,694,707.39 ± 3,002,149.98
Nucleotides	2’-O-Methyladenosine	362,280.31 ± 171,861.23	121,738,612.1 ± 15,919,248.33	89,000,086.69 ± 9,973,817.29	98,214,445.33 ± 11,808,605.37	58,592,631.05 ± 5,785,864.66	25,330,029.91 ± 5,121,593
Alkaloids	Crotonoside	769,632.02 ± 171,685.55	13,377,892.58 ± 2,713,842.41	37,705,047.17 ± 3,139,654.42	99,257,029.27 ± 8,429,231.15	50,940,303.36 ± 5,075,677.01	27,310,409.98 ± 3,017,556.38
Carboxylic acids	D-Glutamine	15,534,750.58 ± 1,117,706.63	22,766,130.05 ± 1,690,608.93	27,069,147.76 ± 4,471,388.18	38,305,543.74 ± 12,664,939.83	5,927,226.67 ± 81,5294.35	10,829,465.34 ± 1,061,020.55
Category	Compounds	A1, A2, A3, A4, A5, A6 Two-by-two comparison of shared differential metabolites					
Phenols	Trimethoprim	170,392,569.41 ± 7,239,534.72	36,939,619.52 ± 884,872.69	91,544,508.75 ± 7,804,644.7	58,070,842.15 ± 469,344.95	131,185,043.54 ± 10,643,889.16	17,080,836.34 ± 527,750.76
Alkaloids	Crotonoside	769,632.02 ± 171,685.55	13,377,892.58 ± 2,713,842.41	37,705,047.17 ± 3,139,654.42	99,257,029.27 ± 8,429,231.15	50,940,303.36 ± 5,075,677.01	27,310,409.98 ± 3,017,556.38

## Data Availability

All related data and methods are presented in this paper. Additional inquiries should be addressed to the corresponding author.

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
