# Peer review of "Widely Targeted Metabolomics Was Used to Reveal the Differences between Non-Volatile Compounds in Different Wines and Their Associations with Sensory Properties"

_foods, 2023, doi:10.3390/foods12020290_

Round 1

Reviewer 1 Report

In regards to the paper I reviewed. The paper answers the question that the researchers are asking are there metabolite differences within the different types of grapes that they analyzed and the answer to that question is yes.

However, I would expect that based upon the number of different types of analysis that they ran through the course of this project. Prior to reading the paper I already knew they would find a difference between especially if you are analyzing different types of grapes.

Do I feel it answers a gap in the field? Yes and no. Do we need to know what the key metabolite differences are, in a sense yes, but I felt like they used ever single piece of analytical equipment at their disposal to look at all of them. It would have been more interesting if they looked at key metabolites and look at their influence of the fermentation capability of yeast. Or something in relation to that. Not just we found all these differences here they are.

The methods overall were well written and could be easily followed by someone who is familiar with each piece of equipment. The references appeared appropriate based on the paper.

The tables and figures are overall whelming at times. Again going back to the targeting approach that the authors used. I understand the need for complex statistics, but the use of a PCA chart to show key differences in the metabolites was overwhelming and somewhat of a gloss over for me. I understand the story the authors are trying to tell with the PCA plot but again with all the samples I didn't know what to focus on.

Other minor issues:

Line 34: I would type out ROS the first type you use it.

Line 99: I am not sure if the "s" is a typo or not. 

Line 124: I would recommend rewriting this sentence. To be past tense.

Line 170: Another typo.

Author Response

Please see the attachment.Thank you for your hard work, experts!

Reviewer 2 Report

Dear another,

First of all, we would like to thank the authors for their valuable efforts. This study is different and interesting as a subject. In general, the desired goal is also good. However, there are problems with the narration and at some points.

*The narrative of the subject is very disorganized. The effectiveness of the work done with such a narrative weakens and cannot reach the reader. The goal and purpose should be clearly stated. The reader should not get bogged down in the details of what this article wants to tell. It's not the length I want to talk about here. Expression and fluency of the entire article, including the abstract; should be corrected with more understandable, clearer and concise expressions. The goal and the result should be more specific.

*I have some concerns about the word metabolite in particular. I didn't know how accurate it is to use only methanol to extract the metabolites of a product, even wine. This topic is very, very important. I suggest changing the word metabolite. Detailed explanations were made in the pdf file.

*Other opinions are commented on the article in the pdf file. You can see in the attachment.

*In general, the article should be rewritten and corrected in detail. major revision required

Author Response

(The authors gave the same response as above.)

Round 2

Reviewer 2 Report

Dear Editor

The author has made the requested corrections. He also stated the relevant explanations. In my opinion, this paper is acceptable.